# Single-Mode Control and Individual Nanoparticle Detection in the Ultraviolet Region Based on Boron Nitride Microdisk with Whispering Gallery Mode

**DOI:** 10.3390/nano14060501

**Published:** 2024-03-11

**Authors:** Jiaxing Li, Qiang Li, Ransheng Chen, Qifan Zhang, Wannian Fang, Kangkang Liu, Feng Li, Feng Yun

**Affiliations:** 1Key Laboratory of Physical Electronics and Devices for Ministry of Education and Shaanxi Provincial Key Laboratory of Photonics & Information Technology, Xi’an Jiaotong University, Xi’an 710049, China; 1549159852@stu.xjtu.edu.cn (J.L.); felix831204@xjtu.edu.cn (F.L.); 2School of Electronic Science and Engineering, Xi’an Jiaotong University, Xi’an 710049, China; chenransheng@stu.xjtu.edu.cn (R.C.); zqf951005@stu.xjtu.edu.cn (Q.Z.); wannian333@stu.xjtu.edu.cn (W.F.); liukang4142@stu.xjtu.edu.cn (K.L.); fyun2010@mail.xjtu.edu.cn (F.Y.)

**Keywords:** optical microcavities, WGM, ultraviolet, boron nitride, nanoparticle detection

## Abstract

Optical microcavities are known for their strongly enhanced light–matter interactions. Whispering gallery mode (WGM) microresonators have important applications in nonlinear optics, single-mode output, and biosensing. However, there are few studies on resonance modes in the ultraviolet spectrum because most materials with high absorption properties are in the ultraviolet band. In this study, the performance of a microdisk cavity based on boron nitride (BN) was simulated by using the Finite-difference time-domain (FDTD) method. The WGM characteristics of a single BN microdisk with different sizes were obtained, wherein the resonance modes could be regulated from 270 nm to 350 nm; additionally, a single-mode at 301.5 nm is achieved by cascading multiple BN microdisk cavities. Moreover, we found that a BN microdisk with a diameter of 2 μm has a position-independent precise sensitivity for the nanoparticle of 140 nm. This study provides new ideas for optical microcavities to achieve single-mode management and novel coronavirus size screening, such as SARS-CoV-2, in the ultraviolet region.

## 1. Introduction

In the field of photonics, boron nitride (BN) is a promising material with a wide band gap (about 6 eV) [1], which provides a transparent optical platform in the ultraviolet (UV) to near-infrared (NIR) range. It is an excellent candidate for UV optoelectronics. With the rapid development of on-chip photonics, micro light sources represented by kinds of microlasers are essential. Compared to visible and infrared light sources, UV light sources have shorter wavelengths, higher frequencies, and stronger absorption of biomolecules, which means they have higher resolution in observing and sensing nano-sized biomatter [2]. Meanwhile, miniature lasers in the form of micro and submicron sized optical microcavities offer the possibility of miniaturization of on-chip light sources. At present, the microlasers are still dominated by ZnO single-mode microcavity lasers and various doped semiconductor lasers [3,4]. In contrast to mature semiconductor materials, optical microcavity fabrication techniques for BN operating in the UV range have not been fully developed, such as whether the thin film layer is robust and can withstand the nanofabrication process without being destroyed. In addition, the undercut techniques used for suspension structure construction and bulk angle etching processes have not been successfully applied to BN [5]. Recently, researchers have studied the hBN photonic crystal microcavity [6] and the composite optical microcavity of hBN and SiO_2_ [7], but there is still a research gap in the pure BN microdisk and microring optical microcavity with suspended structure. Management of single-mode laser and nanoscale sensing by optical microcavities in UV has been a great challenge in micro ultraviolet laser and high-sensitivity nano detection.

The whispering gallery mode (WGM) optical microcavity has attracted much attention in quantum electrodynamics [8], low-threshold lasers [9], and biological sensitive detection [10], because of its high-quality factor (Q factor), exceptionally small mode volume, and heightened sensitivity to the dielectric environment. Nevertheless, the above applications often require the cavity to have a single-mode output with an adjustable wavelength. At present, two methods are available for mode selection in WGM optical microcavities. One is to reduce the size of the microcavity and expand the free spectral range (FSR) to obtain single-mode selection. However, the obtained Q factor of the single-mode by this method is extremely low (about 10^3^) [11,12], which is not suitable for the preparation and integration of high-efficiency sensor devices. The other is to use the Vernier effect to prepare coupled WGM microcavities. The selection of a resonant mode is achieved by adjusting the size of the microcavity and the coupling distance [12,13,14,15,16]. Its advantage is that it can maintain a single-mode with a high Q factor. However, most of the previous research is based on optical microcavities that resonate in visible or even NIR regions. This greatly limits the development and application of micro UV lasers.

The WGM optical microcavity is highly desirable for nanoscale object detection in various fields, such as early disease diagnosis and environmental monitoring [17]. The WGM optical microcavity sensing mechanism mainly depends on the strong interaction between light and matter. The negligible perturbation can lead to faint changes in the mode field, which will affect the significant changes in the resonance characteristics. The mode shift [18], mode splitting [19], and mode broadening [20] were always used to determine the sensitivity of microcavities. When detecting an elastic scatterer, since no absorption loss is introduced, its backscattering will destroy the degeneracy of WGM, and the coupling between propagation modes leads to the shift and splitting of the resonator modes. However, when the mode splitting or shift is smaller than the mode linewidth, it cannot be resolved. This case requires a very high Q factor to improve the resolution. In contrast, when the nanoparticles are absorbent, both the absorption properties of the nanoparticles and side scattering lead to a loss of resonance energy, which in turn leads to an increase in line width. The mode broadening mechanism has the advantage of natural resistance to environmental interference, without ultra-high Q factors. Yang et al. have confirmed the feasibility of the mode-splitting sensing mechanism by detecting polystyrene (PS) particles [19]. Xiao et al. also verified the viability of the mode-broadening detection mechanism by detecting PS particles and single virus particles [20]. Furthermore, researchers use Gires–Tournois Immunoassay Platform (GTIP) to realize label-free bright field visualization imaging and quantitative measurement of biological particles [21]; in addition, using nanophotonics to achieve on-chip integrated refractive imaging and biodetection, combined with machine learning for data analysis and device design, will also have a profound impact on advancing global healthcare technology in the future [22].

In this paper, we designed the BN-based optical microdisk cavity through Finite-difference time-domain (FDTD) numerical simulation. The WGM characteristics of a single BN microdisk cavity with different sizes were investigated, realizing the WGM from 270 nm to 350 nm. Additionally, the cascaded coupling models of multiple BN microdisk cavities for single-mode control at 301.5 nm were explored via the Vernier effect. The single PS nanoparticle size sensing properties of a 2 μm BN microdisk for 140 nm and the single-mode control and individual particular nanoparticle size screening of the WGM mode in the UV band were realized, which has great application potential in UV single-mode microlasers and biological sensitive detection.

## 2. Structure and Methods

Figure 1a shows the structure of a BN microdisk cavity with a thickness of 200 nm, supported by a silicon pillar. At present, the BN film growth technology has been relatively mature, such as the use of RF-Sputtering technology for large-area film growth [23] and Low Pressure Chemical Vapor Deposition (LPCVD) on the high-quality epitaxial technology of different substrates [24]. Etching of BN was performed with an inductively coupled plasma-reactive ion etching (ICP-RIE) process using SF_6_ plasma [25] and e-beam lithography (EBL) [26] to provide technical conditions for graphic BN film. For substrate undercutting, taking silicon substrate as an example, it can be achieved by wet etching with a mixed solution of hydrofluoric acid and nitric acid [27], or by selecting XeF_2_ specifically for isotropic selective removal of silicon [28]. Figure 1b shows the complex refractive index (*n*, *k*) of BN. In the wavelength band above 263 nm, the real part’s complex refractive index *n* is larger than the environment (*n*_air_ = 1), and the extinction coefficient *k* is extremely small. Meanwhile, as shown in the inset in Figure 1b, the BN only absorbs strongly in the extreme ultraviolet band below 263 nm. These provide the conditions for the BN-based optical microcavity to realize the UV WGM modes.

The FDTD method was carried out to study the resonance mode and field intensity distribution of the BN microcavity. Ansys Lumerical FDTD Solutions software (Ansys, Vancouver, BC, Canada, https://www.ansys.com) was used for this work. For optical microcavities, such as microspheres, microtubes, and microdisks, the WGM is mainly located at the edge of the equatorial cross-section of the cavity [29]. Firstly, we simulated the WGM characteristics of a 4 μm diameter BN microdisk cavity using 3D-FDTD simulations. Since the extremely thin thickness of the microdisk cavity provides a field limitation in the vertical direction, for the convenience of the study, we subsequently performed 2D-FDTD simulations using the equivalent refractive index method, with the ambient background medium as air (*n* = 1). The boundary conditions in the X-axis and Y-axis directions were set to the perfectly matched layers (PMLs). A TE mode dipole source with a bandwidth of 180 nm to 400 nm was added to the edge area of the structure to excite all of the optical modes in the microcavity. In 2D-FDTD simulations, the electric field monitor was installed to acquire the electric field distribution map in the center cross-section of the microdisk (XY plane), and the time monitor was situated at the microdisk’s edge to obtain the electric field resonance mode.

## 3. Results

### 3.1. WGM Mechanism of Single BN Microdisk

Figure 2 shows the resonance spectrum of the WGM and the electric field distribution characteristics in different directions of a 4 μm BN microdisk by using 3D-FDTD simulation. In Figure 2a, there are two parts of the region; the purple region represents the UV-visible absorption area of the BN material (black solid line), and the orange region represents the WGM resonance mode spectrum (red solid line) in the BN microdisk. According to Figure 1b, the absorption cutoff sideband of the BN thin film was 263 nm, while the WGM in the BN microdisk cavity ranged from 270 nm to 350 nm, belonging to the deep UV region. This also verified that the BN material provided a wider optical window for the deep UV band WGM resonance. Figure 2b represents the electric field intensity distribution diagram in the XZ plane at the resonance wavelength of 300 nm, where the white rectangular box is the edge of the microdisk cavity. The electric field was mainly distributed at the edge of the microdisk cavity, and a small amount of energy was leaked to the surrounding environment on both the upper and lower surfaces of the microdisk edge. The XY plane electric field profile is shown in Figure 2c, where the field was distributed along the edge of the BN circular cavity. Similarly, in Figure 2d, most of the energy of the electric field was concentrated within 500 nm from the edge of the cavity, and a small part of the energy diffused into the air in the form of an evanescent field with a diffusion length of about 200 nm.

In WGM resonators, optical resonance was the result of repeated cycles of total internal reflection (TIR) of photons in the cavity [30], and TIR occurred at the interface between the high-index materials and the surrounding environment. The prerequisite for realizing WGMs’ resonance was the refractive index of the microcavity material. When the light of a particular wavelength traveled along the sidewall of the cavity for one circle and satisfied the phase-matching requirement, the self-interference effect took place. This resulted in resonant enhancement of the specific wavelength, while others were filtered out. The resonance conditions could be expressed as Equation (1) [31].
(1)$$2\pi Rn_{eff}=m\lambda$$
where *R* is the radius of the resonator, *n_eff_* is the effective refractive index of the resonator, and *m* is the angular quantum number, which is a positive integer, there exists a correlation between resonant wavelength and the size of the cavity.

The free spectral range (FSR) is a crucial parameter in WGM. It is defined as the separation between two adjacent optical modes, such as order *m* and *m* + 1. The FSR varies inversely with the cavity size (*D*) by Equation (2) [31].
(2)$$\mathrm{FSR}=\frac{\lambda^{2}}{\pi n_{eff}D}$$
where *D* is the diameter of the resonator, *λ* is the wavelength maximum of the mode, and *n_eff_* is the effective refractive index of the resonator. The FSR of the resonant mode can be regulated by the varying microcavity size.

The quality factor (Q factor) is an indicator to measure the light trapping capability of the microcavity, expressed as the ratio of the light energy stored in the microcavity to the energy lost per round trip cycle. It can be calculated from the following ratio [32]:(3)$$Q=\frac{\omega }{\Delta \omega }=\omega \tau$$
where the
Δω is the full width at half-maximum (FWHM) and is associated with the considered resonance frequency
ω.

Figure 3a–d show the resonance mode spectra of BN microdisks with diameters of 4, 2, 1.6, and 1 μm by using 2D-FDTD simulation, respectively, and the inset plotted the electric field intensity distribution of the equatorial surface of the microdisk at resonance wavelengths. The black line indicates the ultraviolet absorption spectra of BN. The resonance mode of the BN microdisk cavity was mainly centered at the range of 270 nm to 350 nm, while there was no resonance spectral signal at the absorption region of BN, which exactly proved that BN material provided an optical window for the WGM in a band higher than 263 nm. We calibrated a partial resonance mode, denoted as TE_*m*,*n*_, where the numbers *m* and *n* are related to the electromagnetic field at azimuthal (equatorial) and radial directions [33,34], respectively. Taking the microdisk with a diameter of 4 μm as an example, the modes with resonant wavelengths at 301.2 nm and 299.5 nm could be calibrated as TE_59,1_ and TE_54,2_, and the other modes were calibrated similarly. The electric field distribution pattern revealed a circular, uniform distribution of the mode field at the edge of the microdisk cavity, with energy loss occurring at the interface of the cavity and surrounding air. It was apparent that as the cavity size decreases, the number of resonant peaks gradually reduced in the range of 270 nm to 350 nm. Additionally, the small size of the microdisk cavity would inhibit the generation of higher-order modes and only retain the basic mode.

In Figure 4a, the Q factor raised with the increase in the microdisk cavity diameter from 1 μm to 4 μm. For a cavity with a diameter of 4 μm, the Q value of the resonance wavelength at 301.2 nm was as high as 3283, which was much higher than the value for 1 μm, with only 88. According to the field distribution diagram in Figure 3a,d, the microdisk with a diameter of 1 μm had a greater energy leakage to the free space compared to the 4 μm. In combination with Figure 4b, the FWHM increased from 2.2 nm to 16.4 nm at 341 nm for cavities with diameters of 4 μm and 1 μm (the colored solid line is indicated by the red circle), and the FSR increased from 3.6 nm to 14.4 nm (the black solid line is indicated by the black circle). It is clear in Figure 3a–d that the decrease in the size of the microcavity made the separation between the two neighboring resonant peaks increase. The relationship between the angular quantum number *m* and the effective refractive index *n_eff_* with the resonant wavelength at different microcavity sizes is shown in Figure 4c,d. For the same size, a shorter resonant wavelength corresponded to a larger angular quantum number *m*. From the perspective of geometric optics and Theorem 1, it could be explained that light with a shorter wavelength had more reflection cycles on the side wall of the cavity. The effective refractive index of the optical microcavity mode was an inherent characteristic of the mode dispersion, which was determined by the phase-matching conditions. The large size of the microcavity meant a higher *n_eff_*, which greatly improved the cavity’s ability to limit light. It was further demonstrated that a microcavity with a larger size exhibited a superior quality factor.

In the study of the WGM of a single BN-based microdisk cavity, the resonant mode in the ultraviolet band of 270 nm to 350 nm was identified, and the effects of cavity size on the Q factor, the FSR, the FWHM, the mode number *m*, and *n_eff_* of the microcavity were analyzed. These results provided theoretical support for the exploration of WGM microdisks in the UV region.

### 3.2. BN Microdisks Cascaded for Single-Mode Control

Previously, we have discussed the WGM properties of a single BN microdisk in detail. The reduction of the cavity size would increase the FSR and decrease the number of resonant modes. Combined with suitable gain media, it was expected to realize the micro single-mode laser [35], but this method would suppress the Q factor of the microcavity. Herein, via the Vernier effect, multiple cavity cascaded coupling models of BN microdisks were proposed to realize single-mode.

Figure 5a shows two types of cascade coupling models for BN microdisks. One was double-cascade coupling with a diameter of 2 μm and 1.6 μm, and the other was a triple-cascade model with a microcavity with a diameter of 1.2 μm added based on the previous double disk structure; the coupling distance between two adjacent microdisks is 30 nm. In multiple cascaded systems, a dipole source was set at the edge of the 2 μm microdisk to excite the WGM, and a point electric field monitor was placed at the edge of a smaller diameter microcavity to obtain the coupled WGM resonance mode. In Figure 5c, the WGM of a single BN microdisk with a diameter of 2 μm was taken as a reference, and the WGM resonance spectrum of double- and triple-cascade coupled systems were plotted. There was a parasitic peak at 300.8 nm near the central mode at 301.9 nm for the double-coupled model, while the triple-coupled model had a pure single-mode at 301.5 nm, which satisfied the characteristics of a single-mode. To analyze the mechanism of the above phenomenon, the WGM electric field distributions of the microdisk coupled systems at different resonance wavelengths were monitored, as shown in Figure 5b,d–f. For the double-coupled system, there was a strong and stable WGM at the wavelength of 294.3 nm in the 2 μm microdisk, while it was relatively weak and deformed in the 1.6 μm. In contrast, at the wavelength of 322.2 nm, only 1.6 μm of microdisk existed in the WGM. Figure 5d shows mode distribution in the coupled cavities when both cavities were on resonance at λ = 301.9 nm, corresponding to (*m*, *n*) = (28, 1) and (22, 1) for the big and small cavities, respectively. As expected, the coupled cavity’s resonance was verified by the presence of resonance patterns. The overlap of maximum field positions in both cavities appeared at the coupling region (red dotted line). Similar results have also been obtained for the triple-coupled system at 301.5 nm.

The above results could be attributed to the Vernier effect. In a coupled microdisk cavity system, the resonant modes circulating in each resonator interacted with each other. One cavity could be regarded as a spectral filter of the resonant wavelength of another cavity. Only when the wavelength met the resonant conditions of both microcavities at the same time were some resonant modes enhanced, while others were suppressed. For the WGM present in the cascade system, the intensity and Q factor of the mode decreased, compared to a single microdisk cavity, which was due to the increased optical path of multiple cavities. The Vernier effect suppressed all side modes and high-order modes and retained the master mode, which illustrated a guiding significance for improving spectral purity and stability, and realizing the development of a high side-mode suppression ratio (SMSR) single-mode [36].

### 3.3. BN Microdisk for Nanoparticle Detection

In this part, the effect of PS spherical nanoparticles on the linewidth of the WGM resonance mode was researched. PS was a desired approximation for virus detection because it had a similar refractive index (*n* = 1.49) to common viruses [37]. The individual PS nanoparticle with a radius varying from 10 nm to 200 nm was placed on the sidewall of the BN microdisk and at the antinodes and nodes, respectively, as shown in Figure 6a,b. To avoid detection errors caused by higher-order modes (see Appendix A for details), the changes of the ΔFWHM of the resonant wavelength for TE_27,1_ optical mode at 309 nm were monitored, as shown in Figure 6c,d. In Figure 6c, the ΔFWHM for the attached nanoparticle with 10 nm was significantly smaller (ΔFWHM ~0.03 nm) than that for a nanoparticle with 200 nm (ΔFWHM ~0.29 nm) at the antinodes. When the nanoparticle was located at the nodes, the change in linewidth caused by 10 nm nanoparticles was only 0.02 nm, but the change reached 0.36 nm for the 200 nm nanoparticles. For nanoparticles of the same size, the ΔFWHM caused by different positions in the microdisk was different. In other words, the antinodes and nodes of the resonant mode electric field had different perceptual sensitivity. The size of PS particles affected the WGM’s Q factor because the larger nanoparticles produced greater losses to the cavity mode than the smaller ones, leading to a noticeable rise in the resonant mode’s linewidth.

Figure 7 shows the difference in the perception of linewidth changes caused by nanoparticles of different sizes at the antinodes and the nodes. Obviously, when the size of the nanoparticles was distributed from 10 nm to 40 nm, the nanoparticles were too small to affect the resonant mode electric field regardless of whether the particles were located at the antinodes and the nodes, so the mode linewidth at 309 nm was almost unchanged. When the particle size was between 50 nm and 140 nm, corresponding to the case of nanoparticles of the same size, a more obvious increasing trend of linewidth was found at the antinodes. However, a significant change was only observed at the nodes when the particle size reached more than 100 nm. These results indicated that the location of the antinodes had a more sensitive particle detection ability for nanoparticles located between 50 nm and 140 nm. When the size of the nanoparticles was continuously increased from 140 nm to 200 nm, the change of linewidth at the nodes was higher than that at the antinodes, and the detection sensitivity at the nodes was stronger than that at the antinodes. For 40 nm to 200 nm nanoparticles, there was competition between the antinodes and the nodes in the particle sensing ability. It was worth mentioning that there was a position-independent property only for the 140 nm nanoparticle, with a ΔFWHM of 0.22 nm, which could be detected with stable precision. In addition, the particle detection characteristics of cascading coupled microdisks were also studied (please see the Appendix A), and there was a large difference in the ΔFWHM between the antinodes and the nodes without an intersection point. This particular property of the single microdisk was expected to enable accurate detection of viruses with a common size of 140 nm, such as SARS-CoV-2 [38].

## 4. Conclusions

In conclusion, the WGM of the BN microdisk cavity based on FDTD was studied systematically. The results showed that the BN microdisk cavity had a stable WGM in the ultraviolet band from 270 nm to 350 nm. Based on the Vernier effect, the cascade coupling models with multiple microdisks were proposed, and a single-mode at 301.5 nm was achieved. Then, the effect of individual nanoparticles of different sizes on the different position of the microdisk was simulated via the mode-broadening mechanism, and accurate detection of a single nanoparticle with a size of 140 nm was achieved, which provided a great application prospect for biomolecular detection.

## Figures and Tables

**Figure 1 nanomaterials-14-00501-f001:**
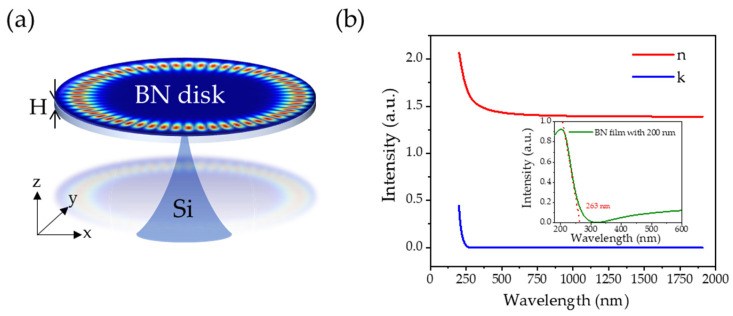
(**a**) The WGM schematic of the BN microdisk cavity. (**b**) Refractive index (*n*) and extinction coefficient (*k*) spectra of the BN. The inset shows the UV-vis absorption spectrum of BN film with 200 nm.

**Figure 2 nanomaterials-14-00501-f002:**
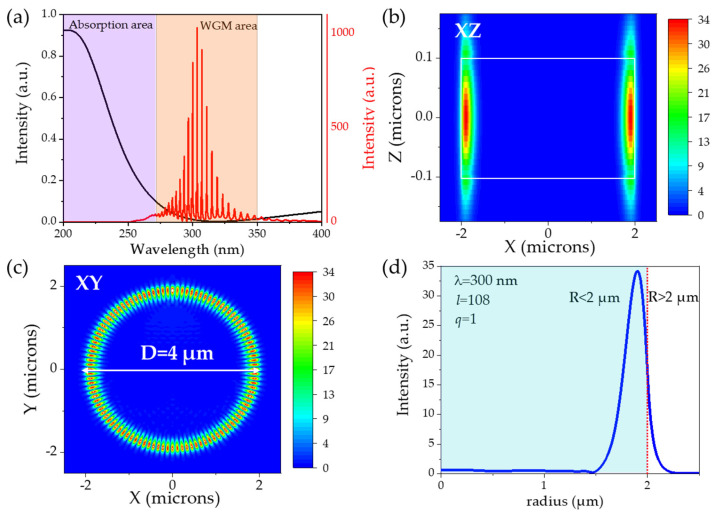
3D-FDTD simulation of the WGM characteristics of a BN microdisk with a diameter of 4 μm. (**a**) The WGM spectrum of the BN microdisk cavity. The electric field distribution (**b**) XZ plane (the solid white line indicates the microdisk edge), (**c**) XY plane, and (**d**) radial distribution at 300 nm, respectively.

**Figure 3 nanomaterials-14-00501-f003:**
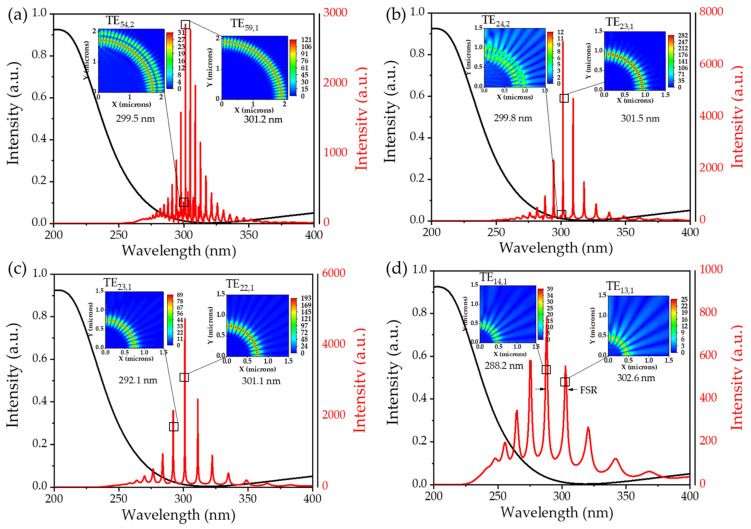
2D-FDTD simulation of the WGM resonance mode spectrum of the BN microdisk cavity with different diameters (**a**) 4 μm, (**b**) 2 μm, (**c**) 1.6 μm, and (**d**) 1 μm. The inset is the XY plane electric field distribution at the corresponding resonant wavelengths.

**Figure 4 nanomaterials-14-00501-f004:**
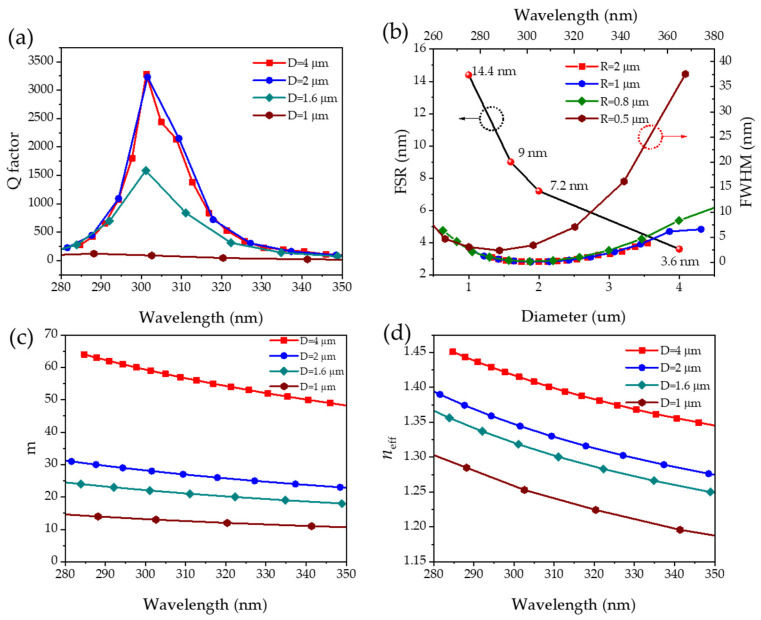
(**a**) The Q factor, (**b**) the FSR and FWHM, (**c**) the azimuthal number *m*, and (**d**) the effective refractive index *n_eff_* for the BN microdisk cavity at different sizes as a function of the wavelength.

**Figure 5 nanomaterials-14-00501-f005:**
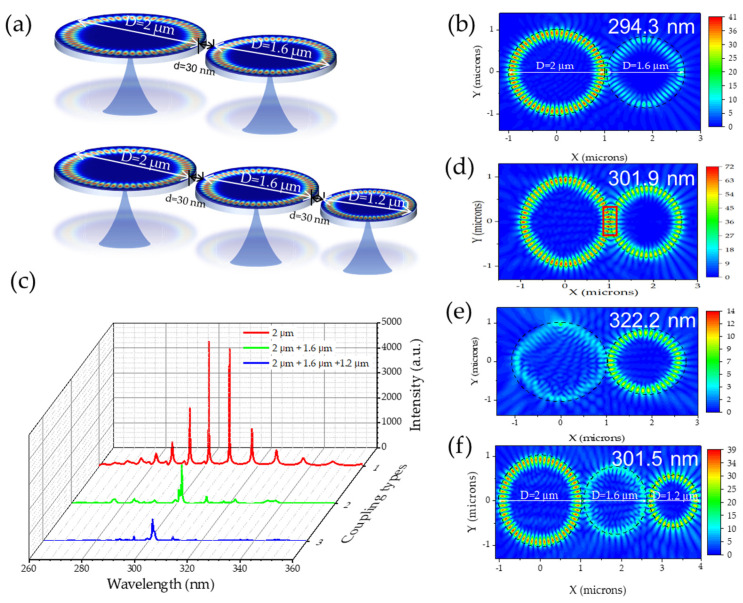
(**a**) Double and triple BN microdisks cascaded coupling models for single-mode regulation. (**c**) The resonant modes spectrum of different coupling types as a function of the wavelengths. (**b**,**d**,**e**) show the electric field distribution of the double microdisks cascaded coupling model at 294.3, 301.9, and 322.2 nm, respectively. (**f**) represents the field distribution of the triple at 301.5 nm.

**Figure 6 nanomaterials-14-00501-f006:**
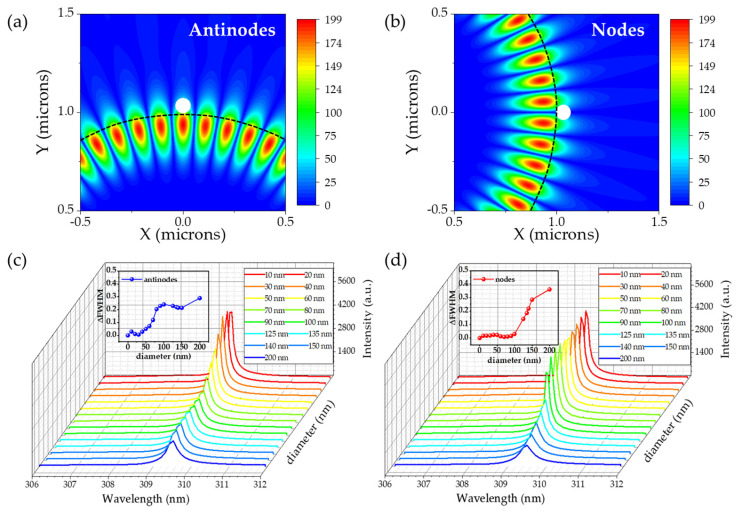
(**a**,**b**) are schematic diagrams of the single PS nanosphere at the antinodes and nodes (the white sphere represents detected PSNP and the black dashed line indicates the microdisk outline), respectively. (**c**,**d**) represent the resonant mode variation with different PS nanoparticle sizes at the antinodes and nodes at 309 nm. The inset shows the trend of the ΔFWHM as a function of the nanosphere’s diameter.

**Figure 7 nanomaterials-14-00501-f007:**
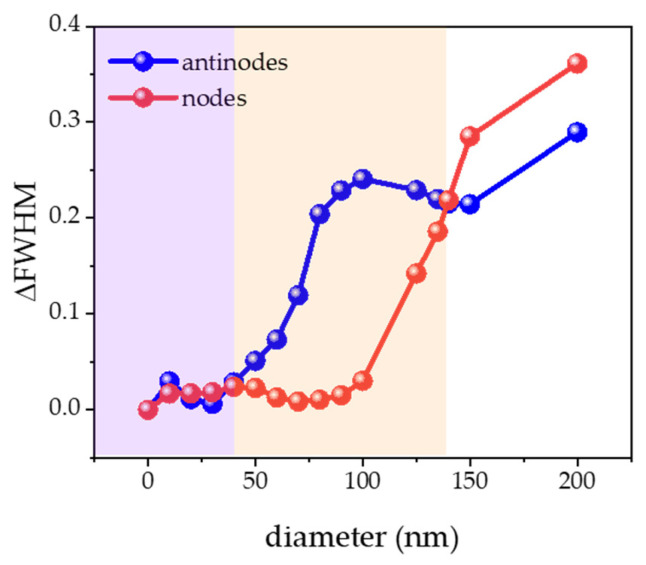
The variation of FWHM (ΔFWHM) as a function of nanospheres with different sizes located at the antinodes and the nodes at 309 nm, respectively.

## Data Availability

The data presented in this study are available upon request from the corresponding author.

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
