# Peer review of "Single-Mode Control and Individual Nanoparticle Detection in the Ultraviolet Region Based on Boron Nitride Microdisk with Whispering Gallery Mode"

_nanomaterials, 2024, doi:10.3390/nano14060501_

Round 1

Reviewer 1 Report

Comments and Suggestions for Authors

Dear Authors, your manuscript is very well written and sound. As a minor revision I would ask to improve the quality of the figures. Figure 3 is quite blurred and the insets are difficult to read. Figure 6, the legends of panels c and d are too small, maybe you should add the description in the caption.

Reviewer 2 Report

Comments and Suggestions for Authors

- In Figures 6a) and b), the position of the microdisc should be sketched as well.

- On Page 5 the texting should be checked "...where the is the full width at half-maximum (FWHM) is associated... ".

Reviewer 3 Report

Comments and Suggestions for Authors

Dear authors,

your article is a very interesting simulation on the possibility of obtaining electromagnetic resonance inside nanometric BN devices therefore in the ultraviolet range.

The analysis starts with the resonance analysis in a disk of 4 micron diameter. Then there is the analysis of disks with different diameters from 4 micron down to 1 micron. The paper finishes with devices composed from coupled devices of different diameters and with analysis of nanoparticle presence on top of the resonant disks.

Apart from some minor misprints present in the paper text, please double check it before publishing, my main question is about the last paragraph of the paper about nanoparticle detection.

Why didn't you use the aforementioned cascaded device ? It would have been much more coherent with the rest of the paper.

Also discussion of Figure 7 regarding particle sensitivity is not very coherent too, since there is the analysis at 309 nm and not at 301 nm, as for the rest of the paper. 

Therefore explanations must be given to the chosen experimental conditions, otherwise last paragraph should be substituted with cascaded filter and PS presence in the R=1 device.

My best regards to the authors

Comments on the Quality of English Language

Dear authors,

some minor misprints present in the paper text, please double check it before publishing,

Reviewer 4 Report

Comments and Suggestions for Authors

This study demonstrated that a BN microdisk with a 2 μm diameter offers precise, position-independent sensitivity for 140 nm nanoparticles, providing new avenues for single-mode management and virus size screening, including SARS-CoV-2, in the ultraviolet region. The manuscript is well-written and timely; however, there are some critical points that need to be addressed before publication.

  • The authors utilized BN-based WGM microresonators, yet, from an engineering perspective, fabricating such structures with BN presents significant challenges. The authors should include a detailed fabrication strategy along with relevant references.
  • The manuscript presents simulation results for detecting nano-sized particles. However, viral particles typically do not exist in isolation but rather in clusters. The authors should discuss this aspect.
  • Moreover, there are recent articles on label-free classification methods for viral particles. The authors need to incorporate appropriate references to highlight the distinctions between their work and existing research, clearly articulating the novelty of their study. (Links: https://onlinelibrary.wiley.com/doi/full/10.1002/lpor.202200814, https://onlinelibrary.wiley.com/doi/full/10.1002/adma.202110003)
Comments on the Quality of English Language

Moderate editing

Round 2

Reviewer 3 Report

Comments and Suggestions for Authors

Dear authors,

I am really unsatisfied of your answer.

I reduced the rating of your overall merit.

I perfectly understood the paper....

My main question has been completely ignored...

Why did you choose 309 nm for particle detection ?

The best peak at D=2microns in your paper is 301 nm

(see Figure 3b of yor paper)...

I still think this is a big methodological error that is 

easy to be solved in a simulation paper...

At least it would be nice to have both wavelength simulations...

Furthermore...

always using 301 nm (see Fig. 5c of your paper  at least 2 disks green line)

it is possible to have appreciable resonance and monowavelength.

Therefore it is absolutely nice to see particle detection behaviour with 

filtered monoresonance (at least two disks 2microns 1.6microns with particle on the 2 microns disk so to be coherent with the present paper final simulation of Fig. 7)

So I asked two simulation more:

- particle detection at 301 nm 2microns disk

- particle detection at 301 nm (2 microns, 1.6 microns disk couple)

Is it so hard to perform ? I don't hink so...

I won't write that I feel offended by the carelessness of your response because perhaps it was my fault for having explained myself poorly in my review.

In case I apologize for my lack of clarity which made you unable to understand my comments precisely.

My best regards

Reviewer 4 Report

Comments and Suggestions for Authors

The revised version is ready for publication.

Author Response

Thank you very much for your careful review of the manuscript.